# Artificial Intelligence Islamic Architecture (AIIA): What Is Islamic Architecture in the Age of Artificial Intelligence?

Ahmad W. Sukkar [1,*], Mohamed W. Fareed [1], Moohammed Wasim Yahia [1], Emad Mushtaha [1] and Sami Luigi De Giosa [2]

1   Department of Architectural Engineering, College of Engineering, University of Sharjah, Sharjah P.O. Box 27272, United Arab Emirates; u22106700@sharjah.ac.ae (M.W.F.); myahia@sharjah.ac.ae (M.W.Y.); emushtaha@sharjah.ac.ae (E.M.)
2   Department of Fine Art, College of Fine Arts and Design, University of Sharjah, Sharjah P.O. Box 27272, United Arab Emirates; lgiosa@sharjah.ac.ae
*   Correspondence: a.sukkar@sharjah.ac.ae

**Abstract:** Revisiting the long-debated question: "What is Islamic architecture?", this research article aims to explore the identity of "Islamic architecture (IA)" in the context of artificial intelligence (AI) as well as the novel opportunities and cultural challenges associated with applying AI techniques, such as the machine learning of Midjourney in the context of IA. It investigates the impact factors of AI technologies on the understanding and interpretation of traditional Islamic architectural principles, especially architectural design processes. This article employs a quantitative research methodology, including the observation of works of artists and architectural designers appearing in the mass media in light of a literature review and critical analysis of scholarly debates on Islamic architecture, spanning from historical perspectives to contemporary discussions. The article argues for the emergence of a continuous paradigm shift from what is commonly known as "postmodern Islamic architecture" (PMIA) into "artificial intelligence Islamic architecture" (AIIA), as coined by the authors of this article. It identifies the following impact factors of AI on IA: (1) particular requirements and sensitivities, inaccuracies, and biases, (2) human touch, unique craftsmanship, and a deep understanding of cultural issues, (3) regional variation, (4) translation, (5) biases in sources, (6) previously used terms and expressions, and (7) intangible values. The significance of this research in digital heritage lies in the fact that there are no pre-existing theoretical publications on the topic of "Islamic architecture in the age of artificial intelligence", although an extensive set of publications interpreting the question of the definition of Islamic architecture, in general, is found. This article is pivotal in analyzing this heritage-inspired design approach in light of former criticism of the definition of "Islamic architecture", which could benefit both theorists and practitioners. This theoretical article is the first in a series of two sequential articles in the *Buildings* journal; the second (practical) article is an analytical evaluation of the Midjourney architectural virtual lab, defining major current limits in AI-generated representations of Islamic architectural heritage.

**Keywords:** aesthetics; epistemology; computer-aided design (CAD); creative design; design methodology; AI; Midjourney; Islamic architecture; architectural visualization; tangible and intangible heritage

## 1. Introduction

(This theoretical article introduces a discussion carried out in a practical article [1]. Reading the practical article after the theoretical one is recommended to clarify the theoretical concepts further through practical examples; however, each article is self-exploratory and self-contained).

The last couple of years have witnessed a proliferation in the usage and prevalence of AI image generators, such as Stable Diffusion and Midjourney, which have undergone

iterations of refinement and have become popular among internet users [2]. These systems utilize text prompts to instantly produce visual representations of architectural concepts in various futuristic, current, and historical context settings [3]. Text-to-image technology employs neural network machine learning, training AI models on extensive datasets to develop decision-making capabilities and create detailed images based on textual inputs. The process involves utilizing diffusion techniques, where the AI introduces random changes to an initial image until it transforms into a new image aligned with the intended concept. This technology comprises two processes: the first involves recognizing, understanding, and deconstructing concepts using existing images, and the second analyzes these images according to the given prompts. These processes combine through iterative collaboration to create novel images that align with the user's prompt. Technology has demonstrated its potential and efficiency in various fields associated with visual domains like architecture, the visual arts, and cinema [4]. In the architectural design process, the emergence of AI has brought about significant transformations, as designers have used it to generate sophisticated, imaginative, and futuristic designs. Figure 1 demonstrates examples of architectural designs produced by Midjourney.

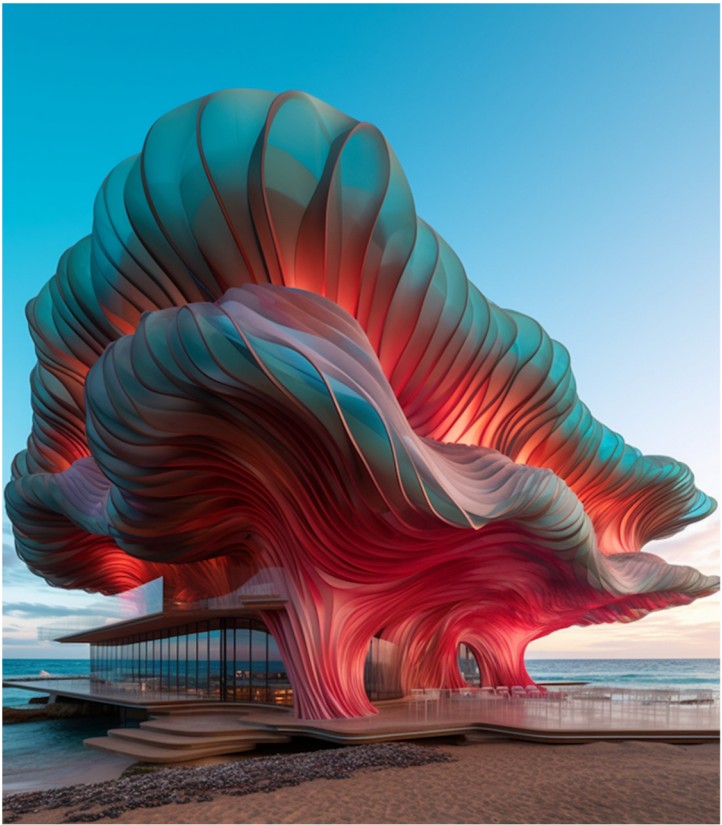

**Figure 1.** An "Instagrammable" example of "blending capacity" in a Midjourney experiment by architect Hassan Ragab [5]. Reproduced with permission.

In a special edition of the journal *Architectural Design*, Campo and Leach [6,7] asked an intriguing question regarding the use of AI as a design method: Can Machines Hallucinate Architecture? They asserted a surging interest in inquiries about the essence of architecture and AI, with a noticeable rise in public curiosity regarding the methodologies involved [7]. "Forget Parametricism and 3D printing", they explain, "the 2020s are all about AI, the first genuinely 21st-century design technique that is revolutionizing architectural culture". Referring to the problem of integrating AI into the unquantifiable creativity, intuition, and sensibility in the design process, they referred to the issues of aesthetics and ethics in using AI in the built environment. Metaphorically, they asked, "Do robots dream of perfect cathedrals?" [7].

Like mainstream global architecture, AI image generators such as Stable Diffusion and Midjourney have been used widely to generate what many designers have described as designs inspired by traditional Islamic architectural styles. However, in culturally, historically, and sometimes "religiously" sensitive fields like "Islamic architecture", approaching image generators through the experimental digital landscape requires critical consideration and a cautious approach. As AI technology evolves, it introduces opportunities and poses challenges for understanding historical Islamic architecture and interpreting its principles through AI's modern architectural design processes. This significant advancement and the rather techno-cultural mutation spearheading it have sparked discussions among users and debates among scholars. Consequently, designers navigating these changes in this critical field have been engaged in a dynamic, ever-changing, and hard-to-be-defined-and-predicted discourse. Figures 2–5 demonstrate examples on the internet and in journal publications. Many examples appearing on the internet demonstrate a mixture of a lack of historical knowledge blended with inadequate datasets.

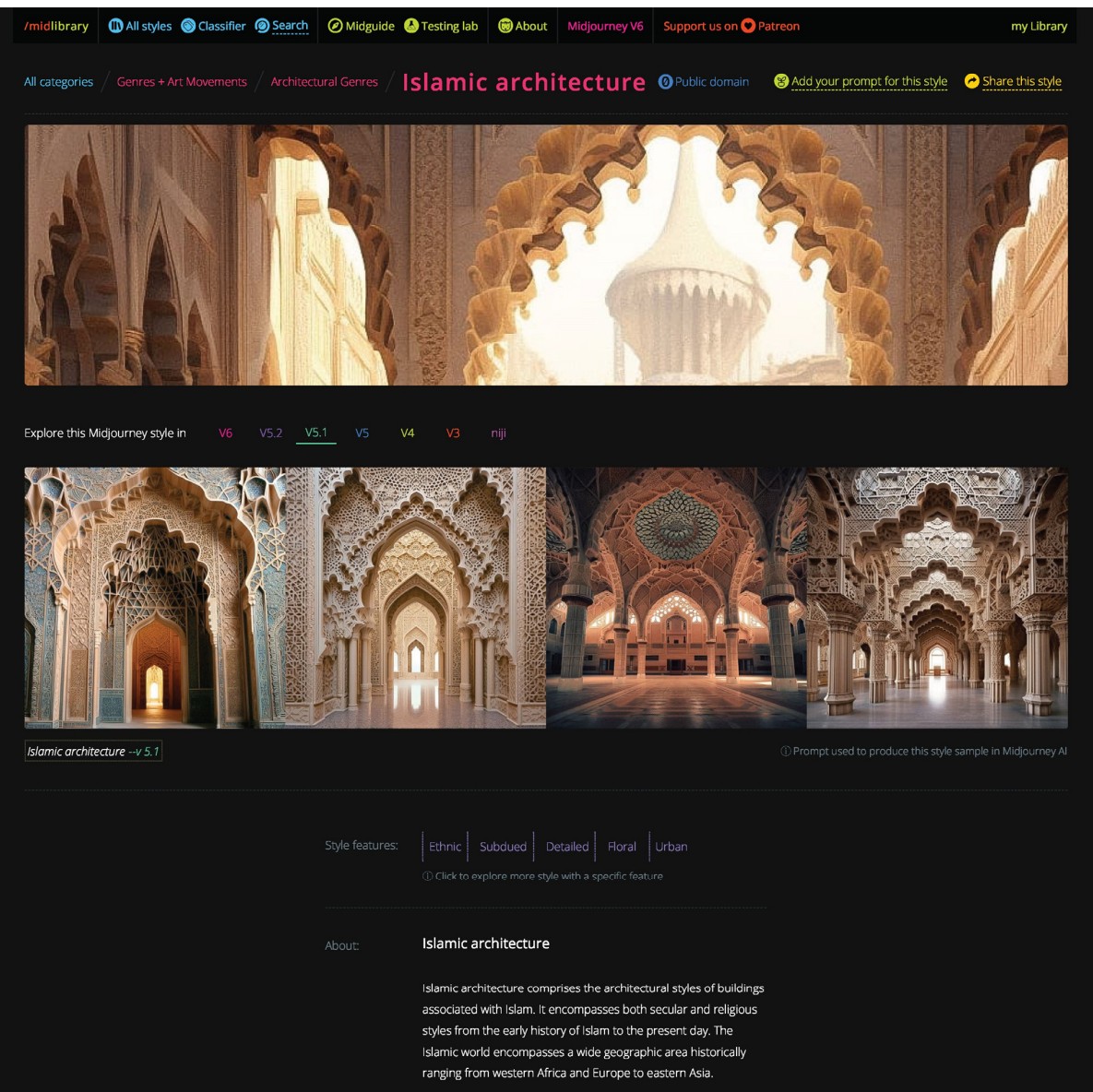

**Figure 2.** Examples of images displayed on open-access online archives generated via Midjourney and classified as "Islamic architecture" [8]. Reproduced with permission.

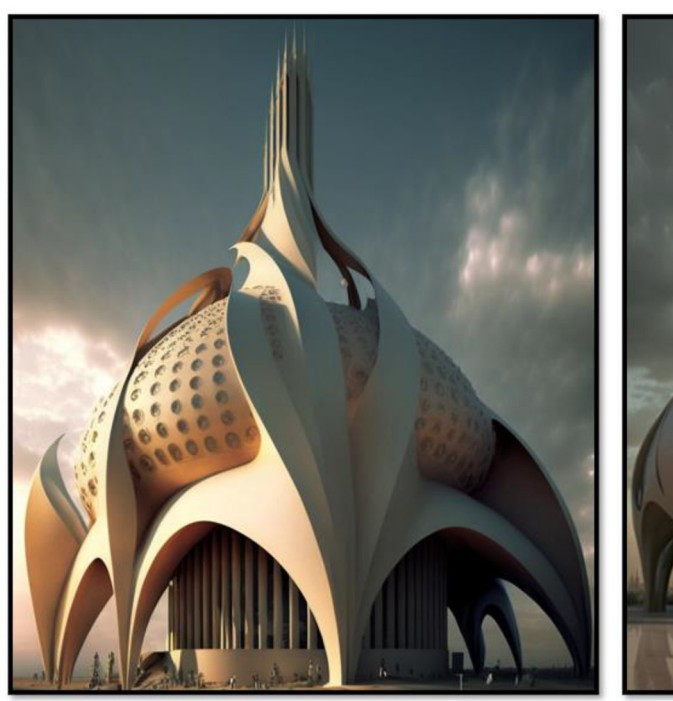
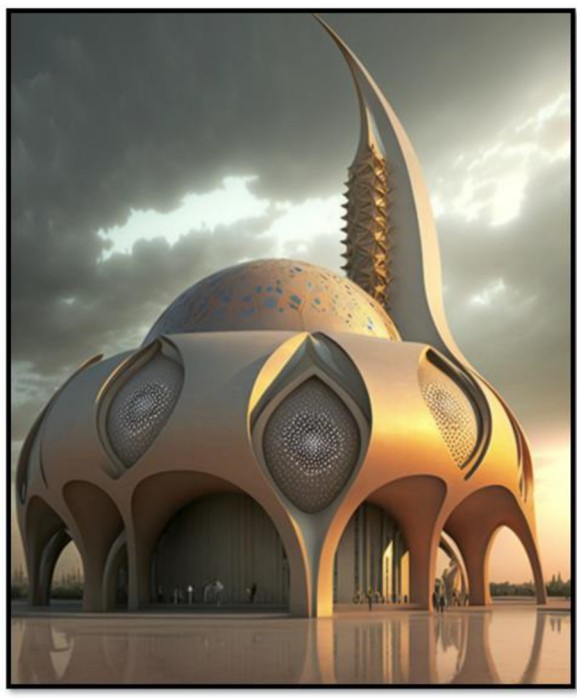

**Figure 3.** Examples that have been described as "innovative designs" featuring "unconventional sculptural architectural façades inspired by the Fatimid architecture ornaments" using Midjourney and Stable Diffusion [9]. Reproduced with permission.

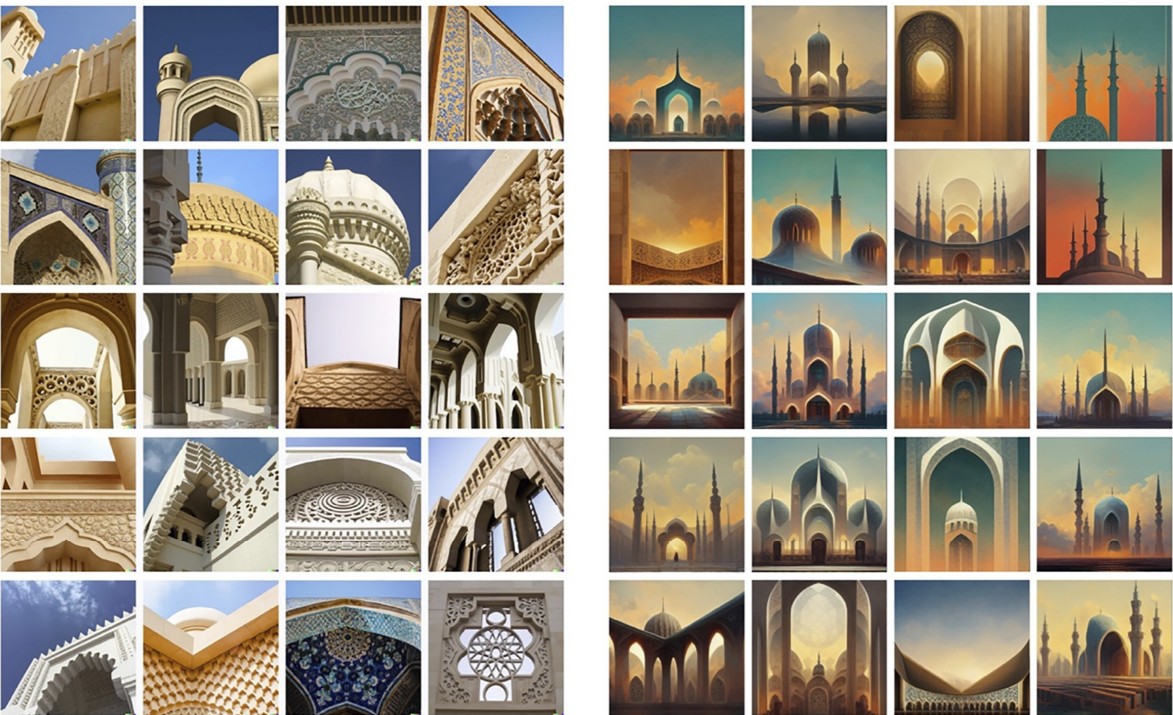

**Figure 4.** Examples of AI-generated images using the prompt "Islamic Architecture". (**Left**): generated by DALL·E 2; (**Right**): by Midjourney, 2022 [10,11]. Reproduced with permission.

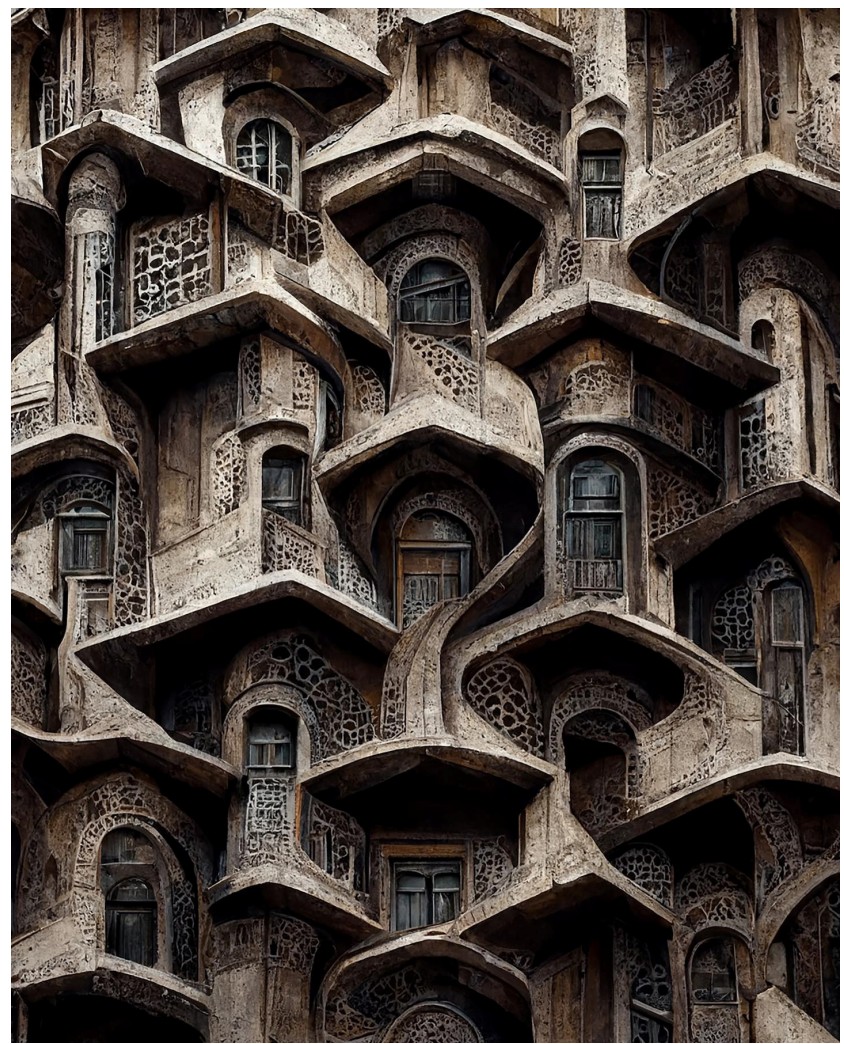

**Figure 5.** An example of the widespread media dissemination of an AI-generated image by architect Hassan Ragab, claimed to be inspired by Islamic geometry and Mamluk buildings, with thousands of views, likes, and shares [12–14]. Reproduced with permission.

The authors of this article argue that this growing field has become a standalone paradigm, termed by them as "Artificial Intelligence Islamic Architecture" (AIIA). This ever-growing organic paradigm is similar to an "organism" with "legs" rooted in history and ethics and arms "stretching" to grasp the future and technology, all through the "body" of modern experimental processes. This article aims to examine this paradigm shift from "Islamicized postmodern architecture" [15] or "postmodern Islamic architecture (PMIA)" [16–19] to "Artificial Intelligence Islamic Architecture (AIIA)". It raises several difficult ethical and aesthetical questions ranging from questioning the appropriate usage of AI in IA to defining the impact of AI on IA. The questions related to IA can be aligned with those raised generally about AI, reaching, at least in science fiction, the extreme of questioning whether AI could cause human annihilation [20–23]. Following these questions, the question of this article revolves around the ultimate question of whether expanding the frontiers of design using AI in IA is a blessing or a curse [24], borrowing a metaphorical expression used in religious contexts. Ultimately, the argument is that AIIA has encapsulated PMIA, defining its boundaries internally and expanding the boundaries of IA externally. Figure 6 symbolically visualizes this boundary root and expansion.

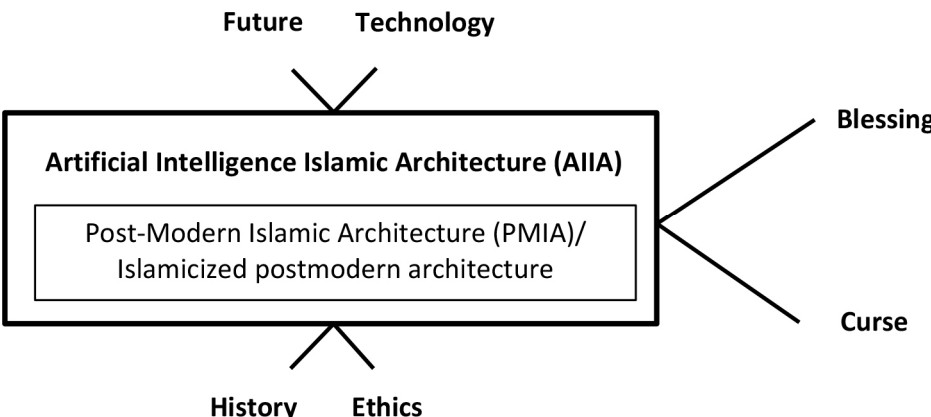

**Figure 6.** Diagrammatic representation of the boundaries of, and crossroads for, artificial intelligence Islamic architecture.

This article suggests a framework to critically initiate thinking about the answers to these questions in a fluid area that is difficult to navigate. It does not answer them conclusively, as the phenomenon of "Artificial Intelligence Islamic Architecture" is still too raw and intangible in architectural materiality to be confronted decisively.

## 2. Literature Review and Research Positioning

Since the age of Orientalism, scholars, especially art and architectural historians, disagree on the definition of "Islamic architecture (IA)". While some have questioned the existence of distinctive Islamic architecture, others have argued that a set of spiritual, socio-cultural, and formal expressions of Islam in the built environment defines it collectively. In modern times, designers have electively used elements of architecture known to be used historically in the Islamic tradition, disassociating them from the holistic structures they were part of in their initial iterations, creating a new architectural hybrid termed by some scholars as "postmodern Islamic architecture" (PMIA) and sparking debates among traditionalists and modernists about symbolism and function in general and about Islamic architecture in specific. In the age of artificial intelligence (AI) and the mass experimental and eclectic approach of the use of machine learning in architectural design to produce 3D-like impressive 2D "fantasy" and "accurate" images, the question "What is Islamic architecture?" arises again, with the need to be reformulated taking into account recent opportunities and challenges in historical, spiritual, and architectural design processes. The revisiting of the identity of Islamic architecture in the age of AI aims to provide insights into the understanding and interpreting of a self-contained branching discourse that the authors of this article identify as "Artificial Intelligence Islamic Architecture" (AIIA).

The current research gap in the exploration of the impact of artificial intelligence on understanding and interpreting modern Islamic architecture design arises from the continuous discussions surrounding the parameters of the definition of Islamic architecture. Despite ongoing scholarly debates for decades, a clear and universally accepted definition of Islamic architecture has eluded the academic community. Though many indirect attempts at discerning the nature of Islamic architecture were made early in the 20th century [25], the first person to try and answer the question "What is Islamic Architecture?" was the historian of Islamic art and the first curator of the Islamic collection at the Metropolitan Museum of Art in New York, Ernst J. Grube [26]. He explored the topic by arguing that Islamic architecture incorporates specific architectural and spatial features inherent in Islam as a cultural phenomenon. On the other hand, Harvard's first Aga Khan Professor of Islamic Art and Architecture and the founding editor of the journal Muqarnas, Oleg Grabar [27], questioned the very existence of distinctive Islamic architecture, suggesting that its nature is difficult to define due to the absence of a unified system of visual symbols in Islamic culture. The head of the Department of Philosophy at Kuwait University, Abdullah al-Jasmi, and Millsaps College's (USA) Professor of Philosophy and expert in the theory

of values, aesthetics, ethics, political philosophy, and philosophy of religion, Michael H. Mitias [28], countered this argument in their article "Does an Islamic Architecture Exist?" by emphasizing the presence of Islamic symbols, such as the mihrab, in mosques, asserting the existence of Islamic architecture. Subsequently, in the article "What is Islamic architecture anyway?" the Aga Khan Professor and Director of the Aga Khan Program for Islamic Architecture at MIT, Nasser Rabbat, proposed a critical historiography of Islamic Architecture [29]. In another article that also employed the historiography of Islamic art and architectural history [15], he built on the question Grube posed in his seminal text [26]. He emphasized the multifaceted nature of Islamic architecture, linking it to history, culture, and the presence of Islam as a formative component within societies, arguing that Islamic architecture is defined by the spiritual, symbolic, social, political, functional, behavioral, and formal expressions of Islam in the built environment. The attempt to consider such perspectives fosters a more inclusive epistemology of the term "Islamic architecture" or less contentious expressions and often favored, such as "architecture in the Islamic world", reducing biases and misrepresentations, including those within the digital context, particularly within the AI design process. Table 1 demonstrates key examples of publications questioning the definition of Islamic architecture towards the question of this article, all aligned with their aims and theoretical approach.

**Table 1.** Examples of publications questioning the definition of Islamic architecture towards the question of this article, as well as their aims and theoretical approach.

| Publication Title | Authors | Aim | Theoretical Approach |
| --- | --- | --- | --- |
| What is Islamic Architecture? | Grube, 1987 [26] | Exploration | Historical analysis and interpretation |
| Reflections on the Study of Islamic Art | Grabar, 2000 [27] | Critical and skeptical | Historicism |
| Does an Islamic Architecture Exist? | Al-Jasmi and Mitias, 2004 [28] | Affirmation against skepticism | Comparative analysis |
| Toward a Critical Historiography of Islamic Architecture | Rabbat, 2008 [29] | Theoretical recapping | Critical historiography |
| What is Islamic architecture anyway? | Rabbat, 2012 [15] | Theoretical recap | Postcolonial analysis |
| **What is Islamic Architecture in the Age of Artificial Intelligence?** | **Sukkar et al., 2024** | **Revisiting continuing ideas** | **Contemporary critical historiography with special regard to the impact of technological advancements** |

Further to situating the theme of this research article in the particular literature on Islamic architecture, it is essential to situate its conceptual epistemology within the broader "evolutionary epistemology", which attempts to explain how knowledge evolves [30–32]. Like recent research on digital architecture that has emphasized the idea of evolution toward united computational and natural ontologies [33], this article attempts to explain heritage and design ontology within Islamic architecture in connection with AI. In this ontology of heritage design, AI-generated images are positioned on a spectrum with two poles: the reality of authentic heritage reproduction and the fantasy of creative design formulation. Figure 7 demonstrates the two poles of heritage design ontology.

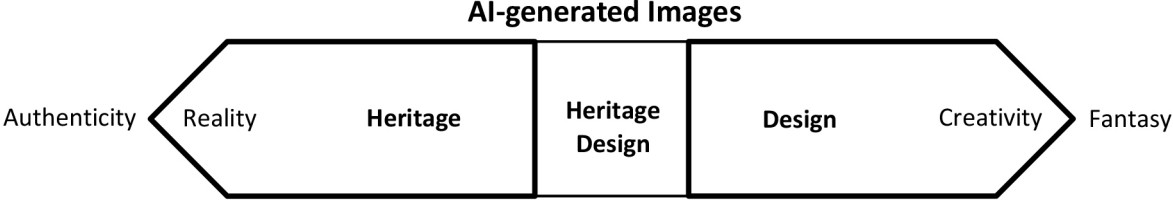

**Figure 7.** AI-generated images in the ontology of heritage design between the reality of authentic heritage reproduction and the fantasy of creative design formulation.

In this broad context of architecture in the Islamic context during the digital age, the significance of this research lies in exploring the controversial identity of this heritage-

inspired design, demonstrating how contemporary design diverges from traditional styles, and explaining the opportunities and challenges of this mechanism.

One of the evolutionary theories related to the central thesis of this article is that of "tacit knowledge", initially developed by the philosopher Michael Polanyi, who coined this influential term [34,35]. Explaining that "we can know more than we can tell", "Polanyi's paradox", to use Autor's expression [36], refers to knowledge learned through experience and internalized unconsciously, which is challenging to codify tangibly. Polanyi's paradox has been widely considered to identify a significant hurdle in AI fields because it highlights the challenge of programming automated tasks and systems without a specific and comprehensive procedure description [37,38]. Experience gained through personal contact and regular interaction with a community of practice is key to acquiring tacit knowledge [39]. Building from evolutionary theories, Tywoniak defines knowledge as rules that reduce environmental uncertainty through connections between ideas and facts, identifying knowledge's four interdependent deformation dimensions as personal, common, tacit, and explicit [40]. Building on Polanyi's theory, Collins differentiates between somatic tacit knowledge, which refers to tasks we do through our bodies but cannot describe how, such as balancing on a bike, and collective tacit knowledge, which is the property of society, such as the rules for language [41]. Numerous studies have explained the importance of the multifaceted concept of tacit knowledge in design [42,43], heritage [44], and architecture [45–47]. However, to date, no studies have examined the effect of AI on IA from the perspective of the concept of tacit knowledge, despite the vitality of this field, given the significant emphasis on hands-on experience, craftsmanship, professional practice, and bodily cognition of architecture in the social and intellectual context of the Islamic tradition [48].

Essential aspects of experience related to the effect of AI on architecture, in general, and Islamic architecture and urbanism, in particular, are those connected to experiential information. De Franco and Moroni defined experiential information in the city's urban context as the information that someone personally absorbs directly by being and acting in a specific urban context; that is, information linked to a "here and now", to a "person on the spot". They demonstrated that experiential information of the mediums of images, sounds, smells, artifacts, and behaviors is often non-linguistic. They further explained that declarative sentences cannot fully express urban experiential information. (In this sense, the knowledge extracted from these experiences is tacit.) They observed that the information the city can provide as an experiential space is crucial even in today's increasingly digitalized world [49–51]. In this context, Islamic architecture and urbanism were lost when they were decoded as an "information system" in the AI-generated digital form; however, aspects of the "experiential input" naturally come through the advanced effect of AI depiction of the physical design of lived-in contexts.

The theories of embodiment, particularly the interdisciplinary field of embodied cognition, offer insights into the issue of the connection between the real world and the digital world, that is, between the body and mind, from both theoretical and practical perspectives. Through wearable technology, sensory environments, and adaptive architecture, Ghandi, Blaisdell, and Ismail demonstrated the possibility of achieving "embodied empathy, using affective computing to incarnate human emotion and cognition in architecture" [52]. Following an experiential approach related to the perspective of a neural network, Lee argues for the importance of a "sustainable embodied experience in the built environment, reinterpreting architectural history through embodied cognition" [53]. Reflecting on atmosphere and memory from the perspective of enactive cognition and neurophenomenology, Pérez-Gómez argued for the importance of "creating life-enhancing atmospheres responsive to human action, embodied emotional memories, and place in the fullest sense (as both natural and cultural context)" through "a proper understanding of consciousness and perception beyond Cartesian misunderstandings" [54]. Several other studies have examined embodied spatial cognition within the framework of neuroscience and architecture [55]. Mallgrave examined several aspects of architecture and embodiment,

illustrating the implications of the new sciences and humanities for design [56]. All these studies show a renewed framework of theoretical implications and practical applications of architecture and embodiment that are potentially applicable in Islamic architecture with regard to its evolutionary knowledge in the age of AI to sustain its significant spatial bodily experience in a human-centric form. The often-deep level of richness of the atmosphere of the produced images in the AI text-to-image model in AIIA would consequently recall questionable embodied memories and emotions.

## 3. Materials and Methods

This research article employs a primarily qualitative, exploratory, and interpretive online desk research methodology based on observation (of mass media, including the internet and social media platforms wherein appear the works of architects, artists, and designers), a literature review (of the relevant published literature), formal analysis of the way AI-generated images related to Islamic architecture are formulated, and ultimately critical historical analyses to contextualize the phenomena of the use of AI within the Islamic architecture field [57]. Examining terminologies, particularly "Islamic architecture", the authors identify fundamental paradigm shifts, stylistic trends, and interdisciplinary controversies in the cross-disciplinary domain of art history (architectural history), Islamic religion, visual heritage, and artificial intelligence. This study focuses on the effects of visualizing data (text-to-image techniques) by creating visual representations of data to assist in understanding complex historical or cultural information within the field of digital humanities.

This article ultimately attempts to evaluate the current transition and shift in paradigm from CAD to AI, the factors that shape this shift, and its effect on this field. In this sense, it is exploratory, particularly as it investigates the nature of Islamic architecture in the age of artificial intelligence. It is a novel academic exploration, as all the extensive publications in the field so far interpret mainly the definition of Islamic architecture before the AI age. Furthermore, the article also contributes to defining this sophisticated field based on observation of current experimental, virtual designs that have the potential to impact construction applications via 3D printing of designs inspired by 2D AI-generated images. Theoretically, it touches upon primary concepts and applications that each require extensive practical research to be conducted. Hence, the approach is flexible and has an open-ended nature.

Given that this research article examines the paradigm shifts in the field of IA in terms of AI in connection with the evolutionary theory of tacit knowledge, it employs and contributes to the methodology of evolutionary epistemology. Although the approach of this research follows critical theoretical and historical methodologies, it is aligned with the experiential research methodologies, particularly the theory of experiential information. The methodological examination of the practical nature of the research materials, including the methods by which AI comprehends Islamic architecture and urbanism in connection with the theories of vision–language matching, adds to the theoretical methods of tacit knowledge. In this context, the examination of the tangible heritage of Islamic architecture and urbanism represented digitally through the text-to-image models as an AI-made "intangible heritage" is grounded in the methodological underpinnings of embodied cognition. It can be argued that the system of embodied cognition of AI-generated images works reversibly in a way where the mind becomes an extension of the body in comprehending the atmosphere of the real bodily lived-in experience of Islamic architecture. In this sense, this research methodology examines the experientiality of representational embodiment, which takes the form of the digital immateriality of AI-generated images, representing an extension of the mind not as the human mind but as AI into the bodily representation of IA AI-generated images. The methodology of this research is also grounded methodically in cultural theory, especially the aspects of memories and identity.

In terms of the research material, the article draws upon the academic literature, historical records, and up-to-date AI-generated images in the mass media, all based on

scholarly debates on the definition of Islamic architecture exemplified earlier. It examines critical works of architectural historians, theorists, and critics. It reflects on the factors affecting the passive understanding and active interpretation of IA in the recent historical process of its definition, which started in the age of Orientalism and has reached a critical historical moment with AI. Table 2 summarizes the research approach and its methods, tools, nature, disciplines, related theories, and target audiences.

**Table 2.** Summary of the methods, disciplines, and aspects of the nature of the used methodology.

| Topic/ Phenomena | Artificial Intelligence Islamic Architecture (AIIA) Opportunities and Challenges | | | | |
|---|---|---|---|---|---|
| **Approach** | Qualitative | | | | |
| **Methods** | Observations | Historical analysis | Contextual analysis | Formal analysis | |
| **Tools** | Critique | New Terminologies | | | |
| **Nature** | Preliminary | Desk research | Online | Cross-disciplinary | Flexible | Open-ended |
| **Disciplines** | Art/Architectural History and Theory | Religion (Islam) | Visual Heritage | Digital humanities | Design | Artificial intelligence |
| **Related Theories** | Evolutionary Theories | Tacit Knowledge | Experiential Information | Embodied Cognition | Cultural Theory | |
| **Target Audiences** | Art/Architectural historians | Cultural critics | Heritage experts | Designers | AI experts and artists | AI common and end users |

## 4. Results

The study differentiates between two main stages of the development of Islamic architecture in terms of modern computer technology: Postmodern Islamic Architecture (PMIA) and Artificial Intelligence Islamic Architecture (AIIA). Although AIIA can be seen as a soft continuation of PMIA, using computational tools that help to achieve accuracy through design standardization and computer-accurate calculation and drawing, AIIA presents significant divergence and development with more advanced AI programs such as Midjourney, creating new opportunities and challenges and often producing more surreal and hallucinating outcomes (Figure 8).

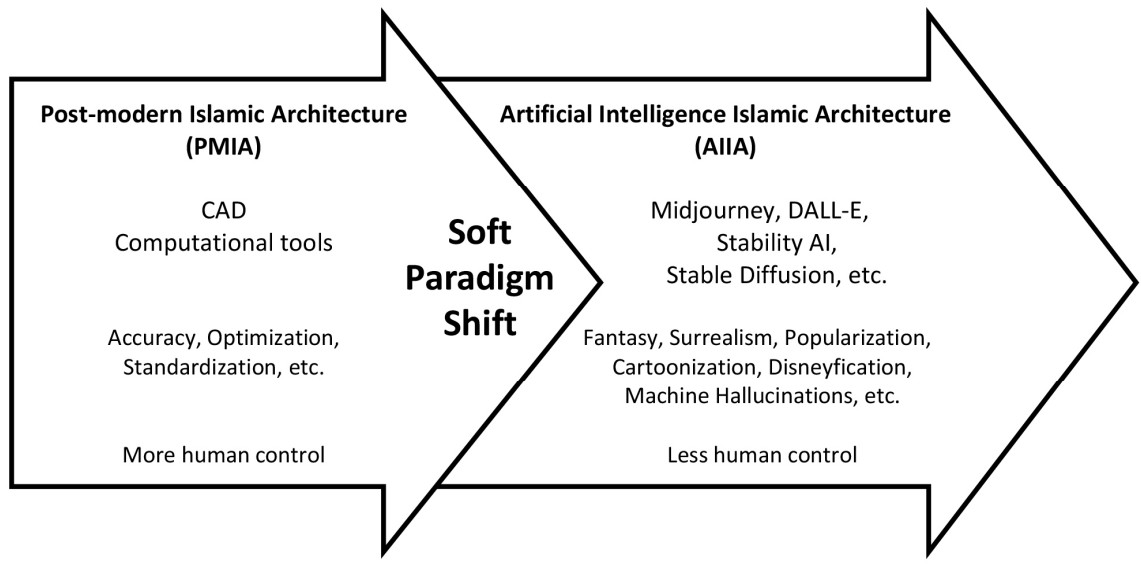

**Figure 8.** A framework of the paradigm shift between two distinct stages in the recent development of Islamic architecture in terms of modern technology.

The study identified the following impact factors of AI on IA, which have shaped this soft paradigm shift in terms of a dramatic change of methods used; in the first, the drawing tools were advanced computationally with a vast space for human control, whereas in the second, a wholly new intelligence became artificial, with more machine control and less human control: (1) particular requirements and sensitivities, inaccuracies, and biases; (2) human touch, unique craftsmanship, and deep understanding of cultural issues; (3) regional variation; (4) translation; (5) biases in sources; (6) previously used terms and expressions; and (7) intangible values.

## 5. Discussion

### 5.1. Pre-AI: Postmodern Islamic Architecture (PMIA)

Before delving into the latest advancements on the impact of AI on IA definition, a historical overview of the factors and milestones that shaped the discourse of IA is essential. To begin with, the terms "Muslim" and "Islamic", used to denote architecture in this context, have distinct meanings, although they are often used interchangeably. The adjective "Muslim" implies a reference to individuals and groups who follow Islam, while "Islamic" describes characteristics, ideas, events, or things associated with Islam [58,59]. While "Muslim architecture" specifically pertains to architectural works produced by individuals who identify as Muslims, "Islamic architecture" encompasses architectural influences from Islam, regardless of the religious background of the creators. In this sense, "Islamic architecture" is an inclusive term, and according to Rabbat [60], Islamic architecture is characterized by its multicultural nature. The term draws inspiration from various sources and engages in dialogue with various architectural traditions. Rather than adhering strictly to a single model or cultural reference, Islamic architecture displays a fusion of influences. Throughout its historical development, creators have adopted, borrowed, and invented ideas, resulting in diverse forms, spatial arrangements, and construction techniques while maintaining a coherent set of intentions and objectives related to Islam as an original religion and applied culture. In a way, the development of this knowledge can be described along Collins' concept of "collective tacit knowledge" [38], where knowledge is framed and produced through a changing network of reference points in society and cannot be formed singularly.

The initial exploration of Islamic architecture, Rabbat [61] explains, was undertaken by European scholars during the colonial period, who collected and interpreted information about architecture in the "Orient". This deep interest of European scholars in the East led to the production of catalogs and records of Islamic monuments, introducing Europe to a previously understudied architectural heritage [62–65]. Said's Orientalism [66] has had a profound indirect impact on subsequent generations of architects and archaeologists, expanding this surveying and classification to cover various regions within the Islamic world [67–69]. However, most scholars neglected the encounter of Islamic architecture with modernity and its implications, especially the evolutions within the field during the nineteenth and twentieth centuries. A Eurocentric perspective influenced post-independence architectural historians in colonized nations, especially the pan-Arabists and Baathists of the 1950s and 1960s, who attempted to reconstruct a "pure" and "authentic" cultural heritage while falling into the conceptual biases of the Eurocentric model. This tendency appears clearly in studies of Central Asian, Iranian, and Turkish architecture, which often prioritize ethnic particularity or national identity over Islamic influences [15,70].

Significant changes occurred in the understanding and interpretation of Islamic architecture with the rise of an ideology emphasizing "Islam" as a defining characteristic [71]. As Rabbat mentioned, Islamic political movements during the 1970s aimed to establish Islamic governance but showed limited interest in the architectural aspects. Simultaneously, the ruling elite in the Gulf region aimed to modernize their cities and establish a distinctive identity, resulting in a demand for contemporary Islamic architecture. Architects responded by integrating "Islamic architectural elements" in their modern, rather Western designs, giving space to a form of architecture known as "Islamicized postmodern architecture" [15].

In the following decades, spanning the 1990s, 2000s, and 2010s, many ambitious architectural projects, often headed by "starchitects", emerged, proclaiming their international style inspired by Islamic architectural elements or motifs [72–74]. Projects such as the Museum of Islamic Art (MIA) in Doha, Qatar, by I.M. Pei (2008), sought to incorporate and reinterpret Islamic architectural motifs in their designs, albeit in diverse and sometimes unconventional manners [75]. The "starchitects" phenomenon of an external culture attempting to integrate their international design values into a local cultural context caused a shift in the Islamic tradition during the last few decades from often being a local "community of tacit knowledge", borrowing Lave's and Wenger's expression [76], to an international or intercultural version sharing a common practice. This "starchitects" phenomenon in several regions of the Islamic world can be described as "Personal Knowledge", which is a kind of tacit knowledge as explained by Polanyi [41], for it involves personality and personal judgment.

Computer-aided design (CAD) advancements have played a significant role in supporting and advancing the movement to redefine Islamic architecture, particularly in affluent Gulf cities like Dubai and Kuwait City. This progress has created iconic landmarks that have become integral parts of city skylines. Notable examples within this ongoing movement include some of the world's tallest neo-futuristic skyscrapers, such as the Burj Al Arab, designed by Tom Wright in 1999, the Burj Khalifa, designed by Adrian Smith in 2010, and the Al Hamra Tower, designed by Gary Paul Haney in 2011. Moreover, further examples in Western and Middle Eastern cities, such as the Institute of the Arab World in Paris and the Louvre Museum in Abu Dhabi, for example, have not only been recognized as architectural wonders but have also played a significant role in defining the unique essence and visual identity of Islamic and Middle Eastern architecture and urban landscapes. These remarkable structures demonstrate the successful integration of advanced technology with traditional (Islamic) architectural elements, both within contemporary settings and internationally. Another example is the Al Bahr Towers in Abu Dhabi, submitted by architect Abulmajid Karanouh from Aedas, which exemplifies the application of advanced technology in redefining Islamic architecture. Completed in 2012, the project draws inspiration from a traditional Islamic element, the mashrabiyya (a wooden lattice screen used in traditional Islamic architecture for privacy, light, environmental control, and natural ventilation). The curtain-wall 150-meter-high office towers feature two circular structures enveloped in a honeycomb-inspired kinetic facade. The design concept earned recognition, winning the 2012 Chicago-based Council on Tall Buildings and Urban Habitat (CTBUH) Innovation Award and was included in its "Innovative 20" list of buildings that challenge the typology of tall buildings in the 21st century [77].

### 5.2. From PMIA to AIIA: A Soft Paradigm Shift

The emergence of advanced digital CAD technology has undoubtedly transformed the field of Islamic architecture, revolutionizing how architects design and construct buildings inspired by Islamic motifs. However, the advent of AI as the fourth industrial revolution [78–80] opened up a new realm of possibilities, inviting scholars and critics to revisit the collective understanding of Islamic architecture in the age of AI.

Recent advancements in text-to-image tools within the architectural domain have laid the foundation for an exciting shift in how Islamic architectural design is conceptualized and created. AI image generators possess the potential to revolutionize Islamic architecture by seamlessly blending Islamic aesthetics, often seen as timeless beauty, especially in the eyes of the (believer) beholders, with cutting-edge technology, all with a click of a button, in a different and often threatening shift of the knowledge formation process, replacing the embedded knowledge of communities with generic and easy-to-create visuals. Despite all restrictions and critical considerations, such a tool has the potential to empower architects, designers, and artists to explore the rich heritage of Islamic architecture in novel and innovative ways.

By harnessing the capabilities of AI image generators, architects can transcend traditional design limitations and explore bold and imaginative interpretations of Islamic architectural elements through the design process. The following sections address the issue of how AI image generators may encourage architects to challenge the limits of representing Islamic architecture in the age of artificial intelligence.

*5.3. Post-AI: Artificial Intelligence Islamic Architecture (AIIA)*

The term "Islamic architecture" is often associated with simplified and stereotypical depictions of ornate domes and arches in the Islamic and Western worlds. Concerning this stereotyping and oversimplification, Rabbat [15] referred to two kinds of students he taught throughout two decades at MIT, who were "a microcosmic—and perhaps faintly comical—reflection of the status of Islamic architecture within both academia and architectural practice today". The first were Muslim students from abroad, Muslim-American students, and Arab-American non-Muslims, who see Islamic architecture as their heritage. The second were those students who perceive Islamic architecture as alluring, enigmatic, and artistically fascinating, evoking a sense of distant and exotic realms. Their exposure to this architecture has primarily been through fictional portrayals like Arabian Nights in previous generations or Disney's Aladdin in recent times, sparking their curiosity and a hint of excitement through fictional representations.

This simplistic perception continues to prevail among the general public despite the extensive publications on the subject, many of which tend to reinforce similar ways of thinking, albeit more grounded. With a specific readership in mind, and in an attempt to popularize the subject, many publications on Islamic architecture focus on a limited historical period, predominantly highlighting monumental mosques, shrines, palaces, and castles rather than other overlooked but unique architectural typologies common in the Islamic world, such as hammams (public baths), bimaristans (hospitals), and everyday architecture. This portrayal tends to present Islamic architecture as a part of the past rather than a dynamic and living tradition. As a result, when attempts are made to incorporate Islamic architectural elements into contemporary designs, they often face skepticism and uncertainty, both in design practice and architectural history. The uncertainties surrounding the definition, extent, and specificity of "Islamic architecture" raise questions that imbue the hesitancy of architectural historians and the superficial manner in which many architects and designers respond to requests for incorporating such elements [61].

This oversimplification of Islamic architecture has continued and even increased in the age of AI image generators, which further amplifies the issue due to their availability, ease of use, and quick workflow. Moreover, the unlimited mass-produced designs derived from biased databases reinforce the prevailing opinion that predominantly simplifies Islamic architecture as synonymous with buildings mainly featuring elements such as domes and arches. This continuation of oversimplified and stereotypical representations undermines the richness and diversity of Islamic architectural traditions.

The issue of bias in text-to-image technology arises mainly from the utilization of existing image pools formed from the overrepresentation of specific architectural styles and digitally generated images based on photography. This bias, Dreith observes [81], leads to a lack of diversity in the dataset, particularly the non-Western architectural dataset of landmarks. Consequently, the technology falls short of encompassing inclusivity and diversity, which poses a significant limitation for users worldwide. In other words, the inability to obtain precise visual representations of specific landmarks hinders the meaningful utilization of technology.

The underlying reason for this limitation lies partially in the generic training dataset of generative artificial intelligence programs and services like Midjourney, which lacks adequate recognition of regional and cultural variations. This limitation raises questions concerning the Islamic tradition, as is the case in similar traditions, exemplified by the problems with AI raised by the Indian architect Radhakrishnan [82], who tried to use Midjourney to visualize Hindu temples that represent his cultural heritage: Is Midjourney AI a

new anti-hero of architectural imagery and creativity? Likewise, Maganga [83] discussed a similar issue when he tried to create a collection of images depicting the vernacular architecture of Africa, and the results he reported were generic huts that misrepresented the richness and variety of African vernacular architecture. The lack of diversity in the generated images mirrors broader issues in the online portrayal of the non-Western world, and the Islamic world is no exception, if not a very special case, given its shared history with Europe. These concerns within architecture as an academic discipline are part of the broader discourse on the creative disciplines of arts, humanities, and science revolving around the question of whether, or to what extent, generative AI is genuinely creative.

Similarly, from limited access to content from non-Western cultures and languages to persistent reductionist narratives, the visual representation of Islamic architecture is often oversimplified. The nuance and richness of Islamic architectural traditions are overlooked in favor of clichéd depictions. Similarly, prompts like "vernacular architecture in the Islamic world" may produce images of mud-brick houses in desert landscapes or Moroccan houses with interior courtyard gardens (*riad*) but fail to capture the immense regional variations in architectural forms and building techniques.

The AI image generators depend mainly on models developed by generative art algorithms, such as Generative Adversarial Networks (GANs), which have proven powerful tools for creating visually stunning and unique artwork. These algorithms often rely on extensive image datasets for training, as the quality and diversity of the training data directly impact the output of the generated art [84]. The biases in publicly available images and their classifications permeate the generated outputs.

Therefore, the images associated with prompts such as "Islamic architecture" or "vernacular architecture in the Islamic world" likely stem from oversimplified image captions from online search engines. Although more specific prompts like "contemporary architecture in Istanbul" can be utilized, these targeted requests tend to yield generalized depictions that perpetuate stereotypes rather than accurately representing the breadth of Islamic architectural traditions. It thus becomes essential to mention that machine learning algorithms often lack an accurate understanding of the global context. As designers, artists, and enthusiasts make profuse use of AI image generators, it becomes vital to critically examine how these speculative images may inadvertently reinforce clichéd representations and misconceptions. The experts should strive to overcome these limitations by fostering a more comprehensive understanding of the diversity within Islamic architecture and challenging the limitations of existing datasets. Alternatively, they could create AI tools, instilling unavoidable "biases" within dataset processes to make general searches related to Islamic architectural prompts more accurate.

More details and further issues on the factors that shape the impact of AI on IA can be categorized under the following interrelated points.

### 5.3.1. Particular Requirements and Sensitivities, Inaccuracies, and Biases

While AI offers advantages such as user-friendliness, accessibility, and mass-production design generation, the technology requires careful consideration. AI tools allow for experimentation and dynamic visions of Islamic architecture, but caution must be exercised to prevent misinterpretations of architectural principles. In heritage studies, AI-generated images offer the opportunity to digitally reconstruct and visualize historical artifacts and landscapes, providing digital archives and immersive experiences. However, concerns arise regarding the accuracy and authenticity of these images, as AI algorithms can cast inaccuracies and even biases if not trained on diverse and reliable sources. Additionally, reliance on AI-generated images should not diminish human creativity and imagination in heritage studies. It is vital to ensure that AI models used for design generation are well-trained using a broad spectrum of authentic and diverse architectural exemplars while considering the specific requirements and sensitivities associated with Islamic architecture. It would be of interest to study possibilities to increase (or achieve) authenticity and reduce bias in such AI-generated images. The question of the elimination of bias looms large in this

respect, especially given the intricate, subjective, and diverse understanding of "Islamic architecture". Furthermore, the concept of bias in the area of AIIA needs to be investigated from the perspectives of the experts and the public at large [85].

### 5.3.2. Human Touch, Unique Craftsmanship, and a Deep Understanding of Cultural Issues

Overreliance on AI-generated designs could lead to a digital architecture lacking the human touch. Architecture, in general, is a multidisciplinary field that encompasses creative human intuition and a profound understanding of historical and cultural contexts. Therefore, establishing criteria for the balance between leveraging AI as a tool and preserving human intelligence in the design process is essential in many contexts. Traditional Islamic architecture, in particular, emphasizes qualitative craftsmanship in using architectural and decorative elements and materials. In contrast, AI image generators may prioritize quantity over quality, undermining the careful craftsmanship and attention to detail integral to Islamic architectural traditions. Therefore, it is paramount to exercise caution to balance mass production and the upholding of esteemed standards of craft and quality associated with Islamic architecture. Furthermore, it is crucial to differentiate between art and architecture. While architecture can be considered a form of art, it often serves specific utilitarian functions. AI image generators often excel in producing visually captivating designs, but they do not deal with functional aspects and are thus incapable of addressing the unique requirements of Islamic architecture. Consequently, human touch, intervention, and expertise are indispensable in evaluating and refining AI-generated designs, ensuring they fulfill architecture's practical and functional needs.

### 5.3.3. Regional Variation

The use of AI image generators in capturing regional variations in Islamic architecture has made significant progress in generating realistic images; however, it still faces drastic limitations in accurately representing the distinctive features of different regions in the Islamic world. Islamic architecture is known for its rich diversity, with each region having its unique style, motifs, and architectural elements influenced by historical, cultural, and geographical contexts. However, AI image generators often struggle to incorporate these regional characteristics into their generated images. For instance, elements like the "mishkat", which refers to diverse manifestations across different cultural contexts, can be misrepresented or overlooked. In Egypt, it predominantly denotes wooden lattice screens that adorn windows, whereas in Turkey, it assumes an elaborate form crafted from stone or marble screens. Conversely, in Morocco, "mishkat" is characterized by geometric patterns skillfully fashioned using plaster or woodworking techniques. Such variations in the physical attributes of "mishkat" exemplify these regions' cultural nuances and distinct artistic expressions. AI image generators might not accurately capture these regional differences because the training on datasets lacks sufficient regional variations and specific architectural details from different Islamic regions. As a result, the generated images may lack the nuanced features that define each region's unique architectural style. Overcoming this limitation requires further research and development to train AI models on diverse datasets encompassing a broader range of regional variations in Islamic architecture. This process would require direct input from researchers through collaborations with software developers by expanding datasets and algorithm optimizations within a global context, considering history, culture, and language differences.

### 5.3.4. Translation

The "lost in translation" issue when using AI image generators is a valid concern, especially considering the diverse languages used in the Islamic world. The meanings of descriptive words can vary across different languages, and this variation can impact the interpretation and uniqueness of the resultant images. Islamic architecture is studied and appreciated by people from various linguistic backgrounds, including Hindi, Turkish, Persian, Hebrew, English, French, and Arabic. Each language has its own descriptive

words and terminology to describe architectural elements, styles, and features. When AI image generators translate or interpret these words, there is a risk of losing some of the nuanced meanings and context. Even within the Arabic language, widely used in the Islamic world, numerous dialects and regional variations exist. These dialects can have vocabulary, grammar, and pronunciation differences, which may impact the understanding and representation of architectural concepts. AI image generators trained on a specific dialect or set of linguistic data may not accurately capture the full range of meanings associated with architectural terms in other dialects or languages. In order to address these challenges, it is crucial to incorporate multi-lingual and culturally diverse datasets when training AI image generators. By including a wide range of languages, dialects, and cultural contexts, the models can learn to generate images that better reflect the unique architectural characteristics and meanings associated with different regions within the Islamic world.

### 5.3.5. Biases in Sources

Using datasets, like Wikipedia, in AI image generators results in heavy reliance on media-sourced images that internet users widely share. This reliance on popular images implies the potential to favor certain architectural styles, notably the Mamluk and Ottoman, while marginalizing others, such as the Fatimid. This approach raises concerns about potential biases and the reinforcement of dominant cultural norms, particularly in regions suffering from civil unrest, religious conflicts, and ethnic divisions, which are frequently depicted in the media. Consequently, the overrepresentation of specific architectural styles in AI-generated images may perpetuate an imbalanced portrayal of Islamic architecture, thereby hindering a comprehensive understanding of its diverse architectural heritage forms and styles within the Islamic world. On the other hand, the limited visibility or exclusion of certain styles can reinforce cultural biases and hinder the representation and thus the appreciation of the architectural heritage of marginalized communities. It can also contribute to the erasure of their history and cultural contributions. Therefore, continuously updating and improving AI models used for design generation can help address biases and inaccuracies over time. Regular retraining of models with new and diverse data sources can enhance the representation of various architectural styles and reduce bias.

### 5.3.6. Previously Used Terms and Expressions

Another aspect to consider is the limited impact of variations in the terms and expressions of Islamic architecture offered by scholars in the past. AI image generators primarily rely on keywords like "Islamic architecture", "architecture in Islam", and "architecture in the Islamic world" to generate results. While these keywords can provide a general understanding of Islamic architecture, they may not encompass the full range of indicative interpretations and definitions scholars provide over time. Standardized keywords may overlook the nuanced and diverse readings of Islamic architecture put forth by different scholars. This can result in a narrower perspective and potentially limit the richness and complexity of the architectural styles encompassed within the term "Islamic architecture". Addressing these issues requires expanding the dataset used by AI image generators to include a broader range of architectural themes, ideas, and styles, including those that may be underrepresented or less popular. This dataset expansion can be achieved through efforts to gather images from diverse sources and regions while also considering the input of local communities, scholars, and architectural experts. By incorporating a more comprehensive dataset, AI image generators can provide a more balanced and culturally responsive representation of Islamic architecture.

### 5.3.7. Intangible Values

According to Alves [86], interpreting universal symbols and archetypes in diverse cultures and religions as part of the collective unconscious relies on cultural, psychological, and anthropological contexts. In the context of AI-generated images driven by algorithms and data, this reliance may be disturbed, as the non-human generative system prioritizes a

formalist and visually centered approach while neglecting the more profound spiritual and symbolic meanings embedded within the design of the sources. In order to overcome this potential limitation, it is essential to consider the holistic approach to AI-generated images in architecture. Making people aware of their tangible and intangible cultural heritage, including myths, dreams, and archetypal images, can be beneficial. Therefore, by incorporating these heritage elements, AI-generated images can contribute to self-knowledge, collective knowledge, and memory, bridging the gap between the unconscious and conscious realms. Future studies could examine the collective effect of AI on architectural taste. Finally, it is crucial to explore how visuals unconsciously stimulate emotions. Architects and AI tools can evoke emotional and spiritual responses through design by understanding the subconscious processing of emotional visual stimuli. This advanced design process can involve incorporating elements beyond the visual realm, aligning with Pallasmaa's emphasis on the multisensory nature of architecture [87]. For instance, an AI-generated image of a mosque featuring intricate geometric patterns could trigger associations with tactile sensations or acoustic ambiance, expanding the sensory engagement beyond the visual sense.

## 6. Conclusions

This article has opened the discussion of AIIA as an interdisciplinary field that explores the sudden impact of artificial intelligence on the tradition of Islamic architecture. This emerging field of the integration of artificial intelligence into Islamic architecture seeks to leverage AI tools and computational approaches to enhance the design, preservation, and understanding of architectural elements inspired by Islamic traditions and their cultural significance. In its advanced and promising form, after addressing the current limitations, AIIA can foster innovation, sustainability, and cultural continuity by combining Islamic architecture's heritage with artificial intelligence's capabilities.

Exploratorily, the discipline of Islamic architecture, or architecture in the Islamic context, has witnessed a soft paradigm shift due to the continuing widespread and public application of computational tools, more recently AI, similar to and in accordance with the mainstream field of architecture. However, AI impacts IA according to the particular historical and cultural sensitivities examined in this article. AI image generators can be a valuable tool in the design process of Islamic architecture; however, it is necessary to use them cautiously. While AI can offer new possibilities and inspiration, it needs to be supported by human expertise and a deep understanding of the principles and traditions of architecture in the Islamic context. The human touch, craftsmanship, and cultural sensitivity integral to architecture in this context should be preserved and not be overlooked. Striking a balance is crucial, and AI should be seen as an advanced tool to augment human creativity rather than replace it entirely. AI-generated images may not capture the intangible aspects of heritage, such as cultural practices, rituals, or oral traditions, as these often go beyond empirical perception and encompass symbolic dimensions. In its current development stage, AI struggles to fully capture the depth and complexity of these elements, which are often intertwined with personal and subjective experiences.

To address these limitations, future studies and advancements in AI models may provide more suitable solutions. Researchers may explore how AI models can be trained on diverse and representative datasets to avoid biases and misinterpretations in the generated designs. Efforts should be made to ensure that the training data reflect the diversity and richness of Islamic architectural traditions from various regions and historical periods. This aim could be achieved by close collaborations between the researchers of Islamic architecture and software developers who work on AI generative tools for dataset optimization and a better correlation between prompts and their visual renders. Moreover, the limitations of AI need to be addressed to capture the craftsmanship, intricacy, and aesthetics of Islamic architectural elements accurately. These limitations require advancements in AI algorithms and techniques that can better understand and reproduce Islamic architecture's unique features and design principles.

It is also pivotal to consider the implications of the widespread adoption of inaccurate AI images of Islamic architecture on the architectural profession, craftsmanship, and education. Architects, designers, and educators should remain critical and mindful of the limitations of AI-generated designs. A reliance on AI without proper human input and expertise can lead to the loss of cultural authenticity and the homogenization of architectural expression. The issues of authenticity and homogenization in blending architectural styles when designing could be further examined in future studies looking at concepts such as historicism, eclecticism, and mannerism in connection with AIIA.

The recognition of oversimplification and biased depictions in Islamic architecture should extend to a more nuanced understanding of the technology, notably Large Language Models (LLMs). These models, while powerful, can inadvertently perpetuate biases embedded in their training data. The challenge lies in mitigating these biases and fostering a more inclusive representation of Islamic architecture. Recognizing the sensitivity of the interplay between technology and representation is vital for refining LLMs and ensuring a more accurate portrayal that aligns with the diversity and complexity inherent in Islamic architectural traditions. Building an AI model tailored explicitly for Islamic architecture in terms of data, computer architecture, and training, taking into consideration statistical concerns and existing biases, is recommended for a more integrated AIIA.

Other potential avenues for further research in connection with AIIA include examining how text-to-image generation for architectural design ideation in cultural contexts can be used in connection with active teaching and learning [88–90] and how to integrate experiential learning in such process of design research within education [91].

**Author Contributions:** Conceptualization, A.W.S. and M.W.F.; methodology, A.W.S., M.W.F. and M.W.Y.; software, A.W.S. and M.W.F.; validation, A.W.S., M.W.F., M.W.Y. and S.L.D.G.; formal analysis, A.W.S., M.W.F., M.W.Y. and S.L.D.G.; investigation, A.W.S. and M.W.F.; resources, A.W.S. and M.W.F.; data curation, A.W.S. and M.W.F.; writing—original draft, A.W.S. and M.W.F.; writing—review and editing, A.W.S., M.W.F., M.W.Y., E.M. and S.L.D.G.; visualization, A.W.S. and M.W.F.; supervision, A.W.S.; project administration, A.W.S. and E.M.; funding acquisition, A.W.S. All authors have read and agreed to the published version of the manuscript.

**Funding:** This research was funded by the University of Sharjah through a Seed research project entitled Islamic Art-ificial Intelligence: Design Impact of Artificial Intelligence on the Perception and Application of Contemporary Islamic Art and Architecture (submitted 2022, accepted June 2023, grant number 2303071011).

**Data Availability Statement:** The data presented in this study are available in the article.

**Acknowledgments:** The authors would like to thank the University of Sharjah for its significant support. Special thanks go to Abdul Wahab Bin Mohammad, Dean of the College of Engineering, and Nadia M. Alhasani, Dean of the College of Fine Arts and Design (CFAD), for their administrative support. Thanks to architect Hassan Ragab, Andrei Kovalev (Midjourney Styles Library), Hatem Tawfik Ahmed, Mostafa Alani, Zuhair Nasar, and Chaham Alalouch (Open House International) for their support with copyright permissions and Osama Hassan for his initial information about Midjourney.

**Conflicts of Interest:** The authors declare no conflicts of interest. The funders had no role in the design of the study, in the collection, analyses, or interpretation of data, in the writing of the manuscript, or in the decision to publish the results.

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
