# Peer review of "Artificial Intelligence Islamic Architecture (AIIA): What Is Islamic Architecture in the Age of Artificial Intelligence?"

_buildings, doi:10.3390/buildings14030781_

Round 1

Reviewer 1 Report

Comments and Suggestions for Authors

The manuscript presents an in-depth exploration of the evolving concept of Islamic Architecture (IA) in the era of Artificial Intelligence (AI), coining the term "Artificial Intelligence Islamic Architecture" (AIIA). It delves into how AI technologies, particularly image generators, are influencing the perception and creation of IA, analyzing historical and contemporary perspectives.

This paper introduces an innovative concept, AIIA, which is a commendable attempt to bridge the gap between traditional Islamic architecture and modern AI technology. However, a clearer definition and depiction of how AIIA differs or evolves from traditional IA would be useful.

A more detailed explanation of how AI technologies were incorporated into the study and the criteria for analysing AI-generated architectural designs would make the study more valuable.

This article presents a new and timely exploration of the role of artificial intelligence in Islamic architecture. With some improvements in conceptual clarity, methodological elaboration and analytical depth, which I have pointed out in my review, this article can make a significant contribution to the field of architectural studies, especially in understanding how modern technologies are reshaping traditional notions of design.

Reviewer 2 Report

Comments and Suggestions for Authors

This work undoubtedly possesses great potential for contributing to the advancement of knowledge in this field. However, there are parts where the article could benefit from some refinement, both in terms of substantive content and formalistic aspects.

Substantive Comments:

In the abstract and introduction, while the authors emphasize the novelty of their research, it may be beneficial to underscore the broader critical value of the study. Consider incorporating references to concepts such as "practical/tacit knowledge," "experiential information," and "embodied cognition" in your investigation. These additions could enhance many of the arguments presented in this work. Bibliography should be expanded.

The issue of over-simplification and biased representations of Islamic architecture is acknowledged, but it could be further linked to the inherent workings of the technology, especially Large Language Models. Addressing statistical concerns, alongside the other issues outlined (from section 4.3.1 onwards), would provide a more comprehensive perspective.

Formalistic Comments:

The language in the article occasionally appears informal or overly empathic. Striving for a consistently formal tone would enhance the scientific soundness the paper.

Ensure that in-text citations conform to the editorial guidelines of the journal. This will contribute to the overall cohesion and adherence to academic standards.

Consider adopting a neutral background for tables and diagrams, reserving colours exclusively for images. This adjustment can contribute to a more visually uniform presentation.

Assess the necessity of all figures, particularly those depicting social media posts. If possible, these could be either avoided or formatted differently to maintain a more focused and cohesive visual narrative.

Thank you for considering these comments, hopefully constructive, and I look forward to seeing the continued development of your valuable work.

Comments on the Quality of English Language

No particular issues detected from the grammatical point of view. 

Reviewer 3 Report

Comments and Suggestions for Authors

This paper explores the place of artificial intelligence on the definition Islamic architecture. Due to nature of the manuscript, paper deals with extensive literature review in one hand. On other hand, the architectural aspects and Islamic perspectives in the context of artificial intelligence. Paper seems to fall within the scope of the special issue titled “Artificial Intelligence and Buildings: Design, Analysis, and Construction”. Based on the content of manuscript, the paper seems to be more like “Review” article, the final decision on the article type is left to the handling editors. According to reviewer there are issues to be solved in the paper:

·       In the paper, there are posts, seems to be taken from Facebook and other related media sites, although these contents are cited or referenced, there some ethical concerns to be solved requiring author permission or copyright permission (which is also related to conflict of interest section)

·       Figure 7 is not clear about what do authors clearly want to state about the framework? The figure may be re-draw to better explain the approach.

·       According to reviewer, the question, which is also the part of the title, the (or potential) answer for “What is Islamic Architecture in the Age of Artificial Intelligence?”, is not clearly solved. There is no specific answers or aspects to be considered or recommendations about this issue? Authors are advised to be more clear in answering this vital question. May be section “Discussion” should be renamed and re-organized in this context.

Comments on the Quality of English Language

Minor editing of English language required. Authors are advised to be careful about punctuation rules and the writing of words.

Reviewer 4 Report

Comments and Suggestions for Authors

The authors investigate Islamic architecture in the age of artificial intelligence. The research topic is interesting, the methodology is appropriate, and the discussion is good. Therefore, my decision is an acceptance with minor revision to be published in “Buildings.”

Here are my comments on improving the manuscript:

- The introduction is too long due to including literature review contents. Please consider separating it into the introduction and literature review sections. After that, please rewrite the introduction concisely.

- The titles of Figures and Tables should be concise. Please update.

- Please remove the links in the titles of the figures (figure 2, 5). Authors can cite and add these links as references in the reference sections based on the Journal format requirements.

Round 2

Reviewer 2 Report

Comments and Suggestions for Authors

The authors have made some effort to incorporate the suggestions that were put forward.

While it's understandable that they may not want to incorporate every suggestion, there are still some important aspects regarding concept selection, literature, and references that are missing.

If the authors have chosen to include concepts like "practical or tacit knowledge" and "experiential information" in their paper, it would be beneficial for them to refer more constructively to the scholars and papers that have introduced and developed these concepts in the academic discourse (evolutionary knowledge, physical design of lived-in contexts).

This suggestion is intended to strengthen the rationale behind the topic and broaden the scope of the article, which could benefit from further enhancement of its critical dimension.

Reviewer 3 Report

Comments and Suggestions for Authors

Authors provided adequate answers to reviwer's questions/comments.

Round 3

Reviewer 2 Report

Comments and Suggestions for Authors

The authors have adequately incorporated and expanded the literature.

As a minor comment, I still believe that a concise (and critical) appraisal of the added literature would be beneficial, particularly in terms of how it aligns with the methods they employ and the concepts they focus on.

A little editing of the text format (especially on Tables) may still be necessary, but overall, the work has sufficiently improved.
